# Grouping promotes both partnership and rivalry with long memory in direct reciprocity

**Yohsuke Murase**[1,2]*, **Seung Ki Baek**[3]

**1** RIKEN Center for Computational Science, Kobe, Japan, **2** Max Planck Research Group 'Dynamics of Social Behavior,' Max Planck Institute for Evolutionary Biology, Plön, Germany, **3** Department of Scientific Computing, Pukyong National University, Busan, Korea

* yohsuke.murase@gmail.com

**Data Availability Statement:** The source code for this study is available at https://github.com/yohm/sim_evo_game_memory3.

**Funding:** Y.M. acknowledges support from Japan Society for the Promotion of Science (JSPS) (JSPS

## Abstract

Biological and social scientists have long been interested in understanding how to reconcile individual and collective interests in the iterated Prisoner's Dilemma. Many effective strategies have been proposed, and they are often categorized into one of two classes, 'partners' and 'rivals.' More recently, another class, 'friendly rivals,' has been identified in longer-memory strategy spaces. Friendly rivals qualify as both partners and rivals: They fully cooperate with themselves, like partners, but never allow their co-players to earn higher payoffs, like rivals. Although they have appealing theoretical properties, it is unclear whether they would emerge in an evolving population because most previous works focus on the memory-one strategy space, where no friendly rival strategy exists. To investigate this issue, we have conducted evolutionary simulations in well-mixed and group-structured populations and compared the evolutionary dynamics between memory-one and longer-memory strategy spaces. In a well-mixed population, the memory length does not make a major difference, and the key factors are the population size and the benefit of cooperation. Friendly rivals play a minor role because being a partner or a rival is often good enough in a given environment. It is in a group-structured population that memory length makes a stark difference: When longer-memory strategies are available, friendly rivals become dominant, and the cooperation level nearly reaches a maximum, even when the benefit of cooperation is so low that cooperation would not be achieved in a well-mixed population. This result highlights the important interaction between group structure and memory lengths that drive the evolution of cooperation.

## Author summary

In the evolution of cooperation, to what extent is cognitive capacity essential? The social brain hypothesis argued that the brain size of primates has increased with the social group size to manage complex social interactions, e.g., to reciprocate cooperation and punish free riders. On the other hand, in the study of the repeated Prisoner's Dilemma, it has been shown that simple strategies that remember only the previous round can unilaterally control the payoffs even against more sophisticated strategies having longer memories. Thus, it is not straightforward to answer the question of how the evolution of cooperation

KAKENHI; Grant no. 21K03362, Grant no. 21KK0247, Grant no. 22H00815). S.K.B. acknowledges support by Basic Science Research Program through the National Research Foundation of Korea (NRF) funded by the Ministry of Education (NRF-2020R1I1A2071670). The funders had no role in study design, data collection and analysis, decision to publish, or preparation of the manuscript.

**Competing interests:** The authors have declared that no competing interests exist.

changes when players are accessible to more elaborate memory-demanding strategies. This paper studies this question through evolutionary simulations and found that longer memory strategies substantially change the picture when the population has an internal structure. This study thus suggests the joint impact between cognitive capacity and the population structure in the evolution of cooperation, although these have often been studied independently.

## Introduction

A game describes interactions among agents that are governed by a set of rules to specify each agent's possible moves and the resulting outcome from the combination of moves [1]. A wide range of social and biological phenomena are thus covered by the theory of games. A successful strategy in a game can often be constructed by requiring certain reasonable properties in a top-down manner. Then the question is whether natural selection can achieve the same goal in a bottom-up way. Sometimes the answer is straightforward: For a symmetric two-person game, if a symmetric strategy profile $(x, x)$ is the unique strict Nash equilibrium, $x$ is evolutionarily stable, and replicator dynamics will converge there. However, this would be the case for relatively simple games. If we can construct a weak Nash equilibrium at best, keeping the evolutionary trajectory close to the equilibrium will be hard. Or, the evolutionary path can be highly nontrivial when the system has multiple equilibria.

Let us consider the iterated Prisoner's Dilemma (IPD) game. It has long been investigated to deepen our understanding of direct reciprocity, one of the fundamental mechanisms to sustain cooperation by means of repeated interactions. Still, the idea that a nontrivial strategy can be derived mathematically by imposing a few requirements is relatively recent: A major breakthrough was the discovery of zero-determinant (ZD) strategies [2], each of which is made to unilaterally enforce a linear relationship between long-term payoffs regardless of the co-player's strategy. An interesting subclass of the ZD strategies consists of 'extortioners,' which guarantee that the player's long-term payoff grows more than the co-player's. However, such extortionate strategies are not favored by selection unless the population size is small enough because they exploit each other so heavily [3, 4]. In contrast, generous ZD strategies make the co-player's payoff higher until mutual cooperation is reached. Those strategies are fairly successful in evolving populations, especially when the mutation rate is moderately high [5]. More importantly, the discovery of ZD strategies has considerably altered our viewpoint on strategic analysis: Recall that a player's payoff depends not only on his or her own strategy but also on the co-player's by the very definition of a game. ZD strategies, on the other hand, enforce restrictions on the payoffs independently of the co-player's strategy and induce an ultimatum on the co-player. The restriction imposed by the strategy is its own invariant properties that can be analyzed, modified, and even designed *a priori* in terms of long-term payoffs.

According to this viewpoint, many well-known strategies are categorized into a couple of classes. Fig 1a shows a schematic diagram of the strategy space, in which generous strategies are overall placed on the left whereas more strict strategies are on the right. First, we have "efficient" strategies, which are depicted as the blue area on the left of the figure. This class is also called "self-cooperators" because each of its member strategies maintains full cooperation when it is used by both players even in the presence of implementation errors [6]. Fig 1b shows the two players' payoffs, and the blue dot indicates their payoffs when they adopt an efficient strategy. For instance, Win-Stay-Lose-Shift (WSLS) players can recover cooperation from erroneous defection, so WSLS is efficient. By contrast, Tit-For-Tat (TFT) players fall into

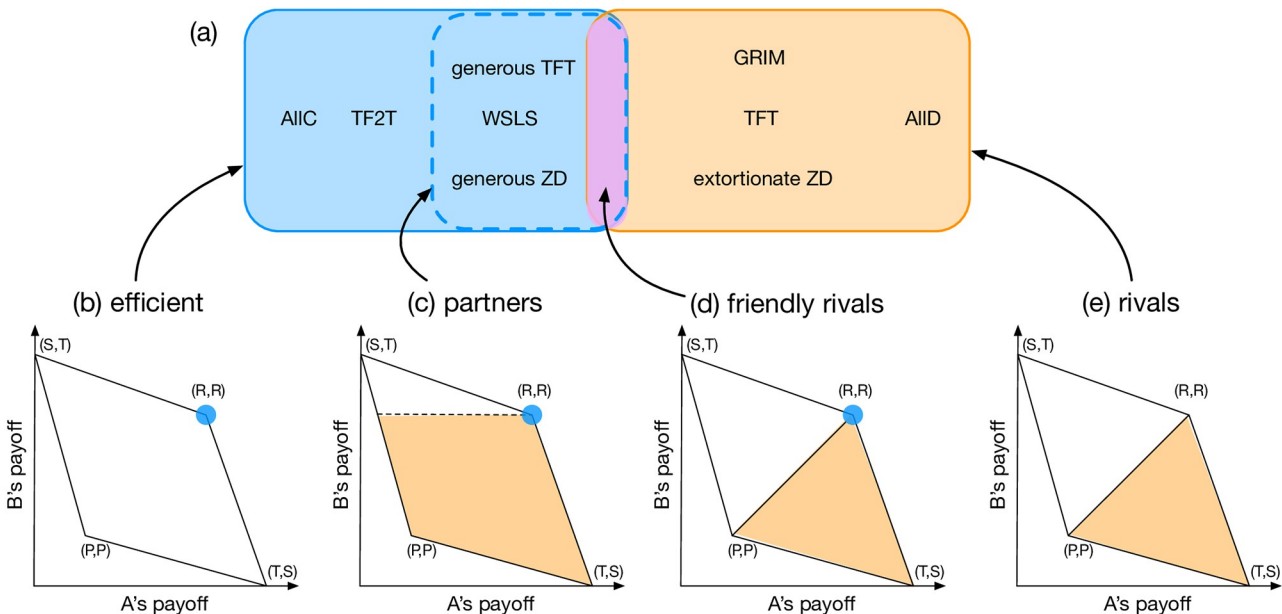

**Fig 1. A schematic diagram of the strategy space.** (a) A schematic diagram of some important strategies showing the four classes of strategies, efficient, partner, friendly rival, and rival strategies. Strategies that tend to cooperate (defect) are shown on the left (right). The bottom panels (b-e) show accessible regions in the payoff space. Each blue dot in (b-d) represents the pair of payoffs when both players use a strategy belonging to the given class. The areas shaded in orange in (c-e) indicate possible payoffs when one of the players, *A*, uses a strategy in the class. The intersection of partners and rivals, indicated by the purple area in (a), defines friendly rivals.

a series of retaliation after a mistake, so TFT is not efficient. The "partners" constitute a subset of efficient strategies, depicted as the area surrounded by the dashed blue square. The partners are also denoted as "good" [7, 8], and all the memory-one partner strategies have been identified [5, 8]. When one of the players, say, Alice, uses a partner strategy, her co-player Bob cannot obtain a payoff greater than the payoff from mutual cooperation, no matter which strategy he takes. It means that Alice unilaterally restricts their payoffs to the shaded area shown in Fig 1c. One of Bob's best responses is taking the same strategy as Alice's to reach full cooperation, which forms a cooperative Nash equilibrium. The other class, rivals, also called "unbeatable" [9] or "defensible," [10] is shown as the red area in Fig 1a. If Alice plays a rival strategy, she never allows her co-player Bob to get a higher payoff than Alice's irrespective of his strategy, thus unilaterally restricting the possible payoff to the shaded area in Fig 1e. This class includes AllD, TFT, and the extortionate ZD strategies.

Based on the above two classes, we can now introduce the idea of friendly rival (FR) strategies [10–14]. These strategies qualify both as partners and as rivals simultaneously, which is indicated as the intersection of partners and rivals in Fig 1a. It achieves full cooperation against itself, and it never allows a lower payoff than the co-player's, as shown in Fig 1d. In this sense, one may regard FRs as the most strict partners, or as self-cooperative rivals. It is straightforward to show that FR strategies are evolutionary robust [5] for any benefit-to-cost ratio and any population size [12]. It has been demonstrated by an evolutionary simulation that one example of FR strategies, called CAPRI, outperforms memory-one strategies overwhelmingly [12]. CAPRI is a pure strategy that behaves like Grim Trigger against most other strategies while it forms full cooperation with itself even under implementation errors. Thus, it is hard for a mutant to invade the community of CAPRI. Nevertheless, the role of FR strategies in the evolution of cooperation remains unclear. FR strategies exist only when memory length is

$m$ = 2 or longer [11], whereas previous studies on longer-memory strategies are relatively limited [15–17] compared to those on the memory-1 strategy space [3–6, 18–26]. Because of the conflicting requirements, FR strategies are quite rare in the longer-memory strategy spaces: the fractions of FR strategies are $1.2 \times 10^{-4}$ and $3.8 \times 10^{-7}$ among memory-2 and memory-3 pure strategies, respectively. Whether these tiny fractions of FR strategies seriously impact the evolutionary dynamics of cooperation is nontrivial.

According to a recent understanding [24], partners are typically selected when the population size $N$ and the benefit of the cooperation $b$ are large, resulting in cooperative states. On the other hand, when $N$ or $b$ is small, a player has a better chance of survival by being spiteful to others [27], which lowers the cooperation level. Therefore, we speculate that the selection of FR strategies is more prominent in an environment where both large- and small-population effects are simultaneously present. In this paper, we consider a group-structured population in addition to the standard well-mixed population of size $N$. In a group-structured population, players are divided into groups and play the IPD game with their in-group members while occasionally imitating strategies of out-group members. The evolutionary dynamics among memory-one strategies in a group-structured population have been studied in detail [28]. We speculate that the group structure plays a more critical role as the memory length increases because FR strategies become available.

In a broader context, we study an interplay among different mechanisms of cooperation [29]. Whereas traditional approaches have focused on characterizing each single mechanism, human cooperation is often ensured by multiple mechanisms working simultaneously, and their interactions are becoming an active area of research. For instance, the joint effect of direct reciprocity and structured populations has been studied intensively [19, 22, 28, 30, 31], and a model unifying direct and indirect reciprocity was proposed [32]. Group structure, in particular, is known to contribute to the emergence of reputation-based norms [33], fairness [34], and kinship structure [35, 36]. This study aims to add another finding to the literature by showing that the underlying tension between inter- and intra-group dynamics induced by the group structure can guide the evolutionary trajectory of direct reciprocity toward the tiny intersection between partners and rivals.

In this paper, we will conduct Monte Carlo simulations of evolutionary dynamics to see the roles of FR strategies in the evolution of cooperation. Specifically, we compare the evolutionary dynamics within the memory-one and memory-three pure-strategy spaces, whose cardinalities are $2^4 = 16$ and $2^{64} \approx 10^{19}$, respectively. (In SI, we show the results for the memory-two strategy space and demonstrate that the results are qualitatively similar to those for the memory-three strategy space. Thus, we focus on the difference between the memory-one and memory-three strategy spaces in the main text.) We will see that a stark difference is observed in the group-structured population and that cooperation approaches the theoretical optimum because of the FR strategies even when the benefit of cooperation is low.

## Model

### Evolutionary dynamics

In this paper, we study the donation game, a special form of the Prisoner's Dilemma (PD) between strategic complementarity and substitutability [37], where the gain from unilateral defection and the loss from unilateral cooperation coincides [38–41]. Its payoff matrix is defined as follows:

$$\begin{pmatrix} b-1 & -1 \\ b & 0 \end{pmatrix},$$
(1)

where the benefit and the cost of the donation are $b$ and 1, respectively. We have normalized the cost of cooperation to 1 without loss of generality. When a donor cooperates, they pay a unit amount of cost, and the co-player gets the benefit of $b > 1$, whereas nothing happens when the donor defects. The benefit of cooperation $b$ is the parameter that controls the strength of the social dilemma. The smaller $b$ is, the more severe the dilemma is. We consider the repeated donation game without discounting the future. Players take unintended actions in each round of the donation game with a small probability $e(> 0)$ because of implementation errors. The long-term payoff of player $X$ against player $Y$ is defined as the average over infinitely many rounds:

$$\pi_{XY} = \lim_{T \to \infty} \frac{1}{T} \sum_{t=1}^{T} \pi_{XY}^{(t)}, \tag{2}$$

where $\pi_{XY}^{(t)}$ is $X$'s payoff against $Y$ in round $t$. When both strategies have finite memory lengths, the sequence of moves is described as a Markov chain. The long-term payoff always converges to a unique stationary value $\pi_{XY}$ and can be calculated by a linear-algebraic calculation. (See Methods for details.)

Players' strategies are updated according to evolutionary dynamics at a longer time scale. Here, we study two types of populations: one is well-mixed, and the other is structured in groups. The well-mixed population, the most standard model in the literature, assumes that each player plays the game with everyone else in the population equally likely. On the other hand, the group-structured population assumes an internal structure in the population so that players play the game only with in-group members while they may imitate out-group members as well. The well-mixed population is a particular case of the group-structured one, so we describe the latter in detail below.

We consider a population of size $NM$, which is subdivided into $M$ groups of size $N$ as shown in Fig 2. Each player plays the repeated game with the other $N − 1$ players in the same group. The fitness of a player $X$ is defined as the average of the long-term payoffs against all the other players in the group:

$$\pi_X = \frac{1}{N-1} \sum_{Y \in \mathcal{G}_X} \pi_{XY}, \tag{3}$$

where $\mathcal{G}_X$ is the set of the other players in the same group.

Suppose that a player currently uses strategy $X$. This focal player is then given a chance to adapt its strategy through either intra-group imitation (with probability $\mu_{\text{in}}$), out-group

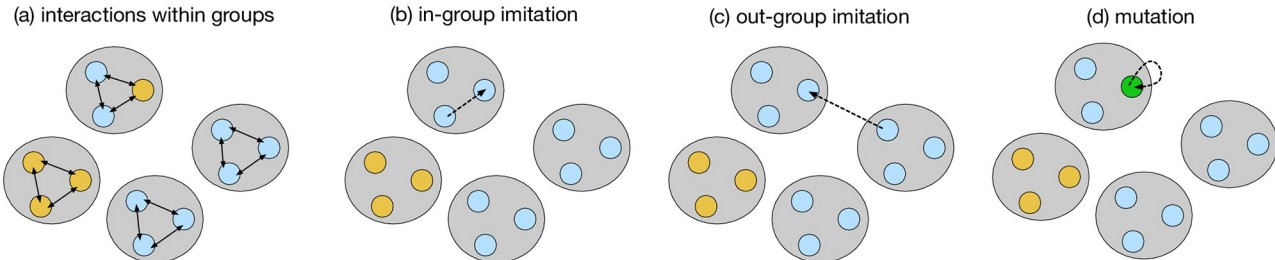

**Fig 2. Evolution in the group-structured population.** A schematic diagram of the group-structured population. In this example, the population is divided into $M = 4$ groups of size $N = 3$. (a) The players in the same group play the game with each other. A player's fitness is determined from the interactions that he or she has been involved in. Each player's strategy is updated by (b) intra-group imitation, (c) out-group imitation, and (d) mutation.

imitation (with probability $\mu_{\text{out}}$), or mutation (with probability $\nu$), where $\mu_{\text{in}} + \mu_{\text{out}} + \nu = 1$. In the case of intra-group imitation, the focal player randomly chooses another player in the group as a role model. If the role model has adopted strategy $Y$, the focal player switches to $Y$ with probability given by the Fermi function

$$f^{\text{in}}_{X \to Y} = \frac{1}{1 + \exp[\sigma_{\text{in}}(\pi_X - \pi_Y)]}, \tag{4}$$

where $\sigma_{\text{in}}$ represents the selection strength of intra-group imitation, and $\pi_X$ and $\pi_Y$ are the fitnesses of the focal and the model players, respectively.

Out-group imitation occurs in a similar way. The focal player randomly chooses a role model from the other groups with equal probability. If the role model uses strategy $Y$, the focal player adopts the strategy with probability

$$f^{\text{out}}_{X \to Y} = \frac{1}{1 + \exp[\sigma_{\text{out}}(\pi_X - \pi_Y)]}, \tag{5}$$

where $\sigma_{\text{out}}$ is the selection strength for out-group imitation. Note that the focal player and the role model are now in different groups so that they do not directly play the game with each other, and this is one of the key differences from a model without group structure. Still, out-group imitation allows strategies to spread from one group to another, just as migration does in genetic evolution models. Finally, when the focal player changes his or her strategy through mutation, the player replaces the current strategy $X$ with $Y$ that is randomly sampled from a given strategy space (see below). This group-structured population reduces to the standard well-mixed one when $M = 1$ and $\mu_{\text{out}} = 0$.

In the following, we assume that intra-group dynamics is faster than both out-group imitation and mutation, whereas the latter two processes have similar time scales, i.e., $\mu_{\text{out}} \ll \mu_{\text{in}}$ and $\nu \ll \mu_{\text{in}}$. In this limit, each group contains two strategies at most: When a new strategy $X$ appears in a group of resident players with $Y$ through either mutation or inter-group imitation, no other mutant strategies will appear until $Y$ takes over the whole group or dies out. The fixation probability of a $Y$-individual in a group of $X$ is then given as [28]

$$\rho_{X \to Y} = \left\{ \sum_{j=0}^{N-1} \exp\left[ \sigma_{\text{in}} j \frac{(2N - j - 3)\pi_{XX} + (j + 1)\pi_{XY} - (2N - j - 1)\pi_{YX} - (j - 1)\pi_{YY}}{2(N - 1)} \right] \right\}^{-1}. \tag{6}$$

Therefore, the probability for a group of $X$ to change its strategy to $Y$ via out-group imitation is given as follows:

$$T_{X \to Y} = f^{\text{out}}_{X \to Y} \rho_{X \to Y}. \tag{7}$$

Let us define relative mutation probability as $r \equiv \nu/(\mu_{\text{out}} + \nu)$, which denotes the frequency of mutation relative to that of out-group imitation. We will see in Results that the ratio between $\mu_{\text{out}}$ and $\nu$ plays a pivotal role in determining an evolutionary trajectory. For completeness, we show the results for an alternative model where the time scales for mutations and out-group imitations are completely separated, i.e., $\nu \ll \mu_{\text{out}}$, in S1 Appendix.

Now we are ready to simulate the time evolution of our model (see Methods for more details). We begin by preparing an initial state with randomly sampled $M$ strategies, one for each group. We define the event of either a mutant or a resident taking over a group as the unit of time. At each time step, we randomly pick a focal group using $X$ among $M$ groups. Out-group imitation occurs with probability $1 - r$: Out of the other $M - 1$ groups, we choose one of them, say, using $Y$. The focal group adopts $Y$ with probability $T_{X \to Y}$. Or, a mutation event occurs with probability $r$, so a mutant strategy $Y$ takes over the focal group with

probability $\rho_{X \to Y}$. The detailed procedure to sample mutants will be explained in the next section. This completes a one-time step, and these steps are repeated until we obtain enough statistics.

## Memory lengths of strategies

Although it is common to define the memory length of a strategy by a single integer, here we define it as a pair of integers $(m_1, m_2)$ to characterize the strategy space in more detail. A memory-$(m_1, m_2)$ strategy prescribes an action based on its own moves over the last $m_1$ rounds and its co-player's moves over the last $m_2$ rounds. For instance, TFT prescribes its action based only on the last move taken by the co-player, so its memory is represented as $(m_1, m_2) = (0, 1)$. So-called "reactive memory-one" strategies belong to this category. As another example, unconditional strategies such as AllC and AllD have $(m_1, m_2) = (0, 0)$. The set of reactive strategies contains unconditional strategies as a subset. However, when we categorize a given strategy in the following, we use the smallest memory length necessary to represent the behavior. For instance, the unconditional strategies belong to the memory-$(0, 0)$ class but not the memory-$(0, 1)$ class. In other words, the set of memory-$(0, 1)$ strategies and the set of memory-$(0, 0)$ strategies are disjoint. See Fig 3a for more examples.

Let $S(m_1, m_2)$ denote the set of strategies that have memory-$(m_1, m_2)$. The memory-$m$ strategy space $\mathcal{S}(m)$ is defined as the set of the strategies satisfying $m_1 \leq m$ and $m_2 \leq m$, i.e., $\mathcal{S}(m) \equiv \bigcup_{m_1=0}^{m} \bigcup_{m_2=0}^{m} S(m_1, m_2)$. The size of memory-$m$ strategy space $|\mathcal{S}(m)|$ equals $2^{2^{2m}}$ because a strategy prescribes either $C$ or $D$ for each of $2^{2m}$ possible memory states. The number of strategies that have exactly memory-$(m_1, m_2)$ is obtained by excluding shorter-memory strategies as

$$|S(m_1, m_2)| = 2^{2^{(m_1+m_2)}} - 2 \times 2^{2^{(m_1+m_2-1)}} + 2^{2^{(m_1+m_2-2)}}. \tag{8}$$

The number of pure strategies for each memory-length pair is shown in Fig 3a. As in Eq (8) and Fig 3a, the number of strategies increases super-exponentially as $(m_1 + m_2)$ grows. If we uniformly sample a strategy from $\mathcal{S}(m)$, strategies with small memory lengths will not appear. For this reason, in order to consider interactions among strategies with different memory lengths, we simulate mutation by using the following two-step process:

$$\begin{cases} \text{Step 1: Sample two independent random numbers } m_1 \text{ and } m_2 \text{ uniformly from } [0, m]. \\ \text{Step 2: A strategy is uniformly sampled from } S(m_1, m_2). \end{cases} \tag{9}$$

This scheme allows us to sample shorter-memory strategies such as AllD, AllC, and TFT with non-negligible probabilities. Furthermore, the average memory lengths would be $(m/2, m/2)$ under neutral selection.

Efficient, rival, and FR strategies are distributed disproportionately in the strategy space. Fig 3b–3d show the fractions of efficient, rival, and FR strategies relative to the whole number of memory-$(m_1, m_2)$ strategies (see Methods for more details). According to Fig 3b and 3c, the fractions of efficient and rival strategies tend to decrease as $m_1$ or $m_2$ increases, although the decreasing trend is milder for efficient strategies. Fig 3d shows that FRs are far rarer than efficient or rival strategies: They exist only when $(m_1 + m_2) \geq 4$, and the fraction goes down to $4 \times 10^{-7}$ in $S(3, 3)$. Thus, the chance of finding an FR strategy through random search is negligibly small.

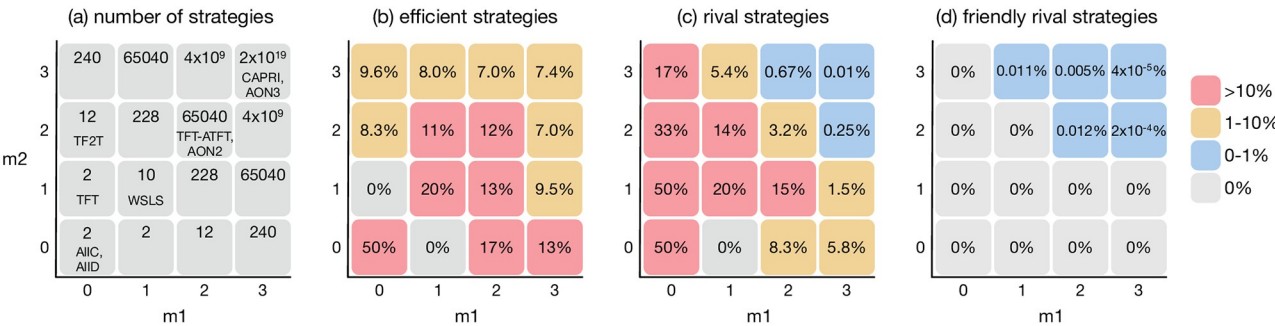

**Fig 3. Numbers of the strategies in each class.** (a) The numbers of pure memory-$(m_1, m_2)$ strategies according to Eq (8). Some well-known strategies in each class are also shown. (b-d) The fractions of efficient, rival, and FR strategies in each memory-length pair. For instance, there are 10 memory-(1, 1) strategies in total. Among them, 20% are efficient strategies, other 20% are rival strategies, and no FRs exist. These fractions are independent of the benefit and the cost of cooperation.

## Results

### Evolution in well-mixed populations

First, we show Monte Carlo results for well-mixed populations in Fig 4. The upper panels (a-d) are the results for $\mathcal{S}(1)$, whereas the lower panels (e-h) are for $\mathcal{S}(3)$. As shown in the figure, both $\mathcal{S}(1)$ and $\mathcal{S}(3)$ show qualitatively similar behavior: When $b$ and $N$ are high, the cooperation level is high, and the population primarily consists of non-FR efficient strategies with few rivals. By contrast, when $b$ or $N$ is small, non-FR rivals occupy most of the population, lowering the cooperation level. One might find it puzzling that memory length is almost irrelevant despite the presence of FRs in $\mathcal{S}(3)$. However, as shown in Fig 4h, FRs actually occupy only a small fraction of $O(10^{-3})$. Although significantly greater than expected from neutral selection,

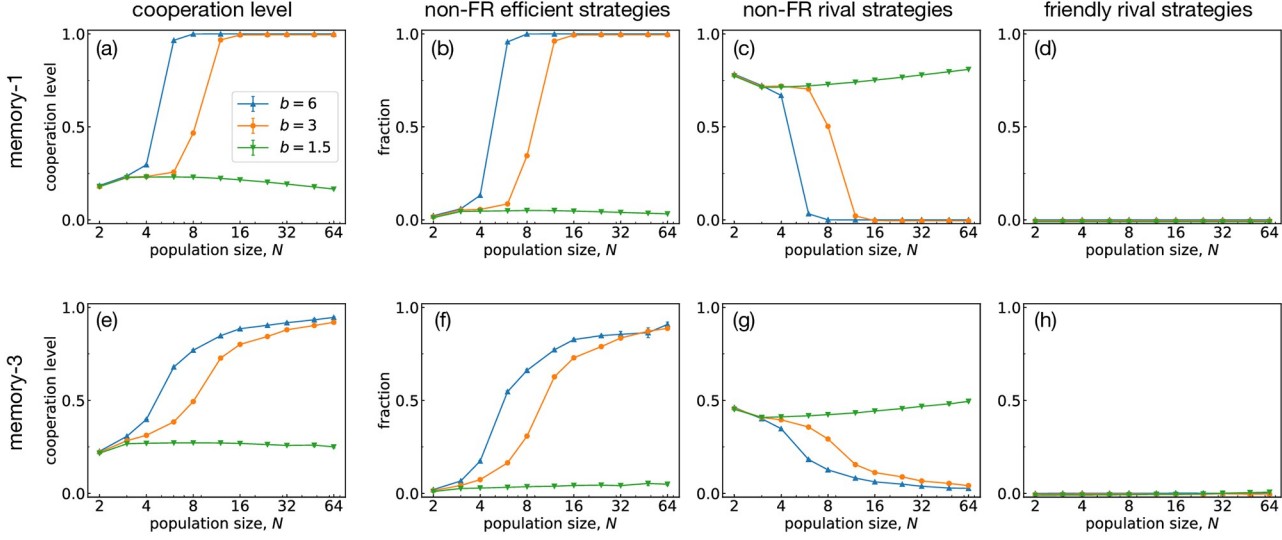

**Fig 4. Evolutionary simulations for the well-mixed populations.** The panels on top (a-d) and bottom (e-h) show the results for $\mathcal{S}(1)$ and $\mathcal{S}(3)$, respectively. From left to right, (a,e) the cooperation levels, (b,f) the fractions of non-FR efficient strategies, (c,g) those of non-FR rival strategies, and (d, h) those of FR strategies are shown as functions of the population size $N$. Throughout this figure, the error probability is $e = 10^{-6}$, and the intra-group selection strength is $\sigma_{\text{in}} = 30/(b-1)$, where the denominator has been introduced to make the typical time scale comparable between different values of $b$. Each data point is averaged over time and over 10 independent runs, after discarding the first initialization period. See Methods for further details.

it is still a negligible fraction compared with non-FR efficient or rival strategies, indicating that FRs play a marginal role in a well-mixed population.

This result shows that cooperation is still challenging for $b = 1.5$ even with $\mathcal{S}(3)$. Although FRs are evolutionarily robust even for $b = 1.5$, other strategies can still replace FRs via neutral drift. In other words, FRs are not successful enough to compensate for their small numbers in $\mathcal{S}(3)$. The same argument also applies to All-or-Nothing-3 (AON3) strategy [17]. AON3 is a memory-3 strategy that forms a subgame perfect equilibrium for $b/c > 4/3$. While $\mathcal{S}(3)$ contains FRs and AON3, it also contains other strategies that can replace them.

One might wonder why the cooperation level for $\mathcal{S}(1)$ is higher than that for $\mathcal{S}(3)$ when $b$ and $N$ are high. For instance, when $b = 6$ and $N = 8$, the cooperation level is approximately 1 for $\mathcal{S}(1)$ while it is around 0.8 for $\mathcal{S}(3)$. This unusually high cooperation level for $\mathcal{S}(1)$ is not because WSLS is exceptional but because there is no dangerous mutant that can threaten WSLS in $\mathcal{S}(1)$. In $\mathcal{S}(1)$, the strategies that can exploit WSLS have low cooperation levels against themselves [28]. However, in $\mathcal{S}(3)$, some strategies, such as CAPRI, can exploit WSLS while having high self-cooperation levels. We confirmed by simulations that WSLS is replaced more easily when we add mutants from $\mathcal{S}(3)$.

Typical time series for these simulations are shown in Fig 5. When we choose $\mathcal{S}(1)$ and $b = 6$, the population adopts WSLS throughout the observation period as expected. For $\mathcal{S}(3)$, efficient strategies are again the majority for most of the time, although we observe frequent turnovers. If the benefit is low ($b = 1.5$), on the other hand, non-FR rival strategies are the majority for both $\mathcal{S}(1)$ and $\mathcal{S}(3)$. One noticeable difference in the latter case is the occasional surges of FR strategies, but they do not last long.

## Evolution in a group-structured population

Next, we show Monte Carlo results for a group-structured population of group size $N = 2$ and the number of groups $M = 10^3$. Fig 6 shows the cooperation level, together with the fractions of strategies categorized into non-FR efficient, non-FR rival, and FR strategies, where the horizontal axis is the relative mutation rate $r$. In $\mathcal{S}(1)$, as shown in Fig 6a–6d, efficient strategies (typically WSLS) and rivals coexist, so the cooperation level is intermediate. This coexistence is observed in a broad range of $r$ and insensitive to $b$, as reported previously [28].

If we consider $\mathcal{S}(3)$, the cooperation level is close to 100% [Fig 6e–6h]. The results again show little dependence on $b$, so a high degree of cooperation is possible even with low $b$. While the cooperation level remains high in a broad range of $r$, the fractions of strategies show nontrivial dependence on $r$: When the relative mutation rate $r$ is higher than $O(10^{-3})$, which is the order of $O(1/M)$, the population is mainly composed of FRs, whereas non-FR efficient strategies replace them as $r$ decreases. When out-group imitation occurs less frequently with small $\sigma_{\text{out}} = 3/(b - 1)$, the pattern changes to some extent in that non-FR rivals are more favored than non-FR efficient strategies for low $r$ [Fig 6i–6l]. Nevertheless, FRs are always the most prevalent as long as $r$ is higher than $O(10^{-3})$. In other words, a high mutation rate helps to promote cooperative behavior supported by FRs.

Typical time series in a group-structured population are shown in Fig 7. For $\mathcal{S}(1)$, non-FR efficient strategies such as WSLS and non-FR rival strategies stably coexist as shown in Fig 7a. This is because of the conflicting requirements between in-group and out-group selection: In-group selection favors a rival to take over the group, whereas efficiency is more important in out-group selection for a strategy to spread across different groups. Fig 8a illustrates what typically happens between WSLS and AllD. Because WSLS is an efficient strategy with a higher payoff, WSLS is more likely to be imitated by AllD players via out-group imitations. However, the newly appeared WSLS player cannot spread in the group because WSLS is weak against

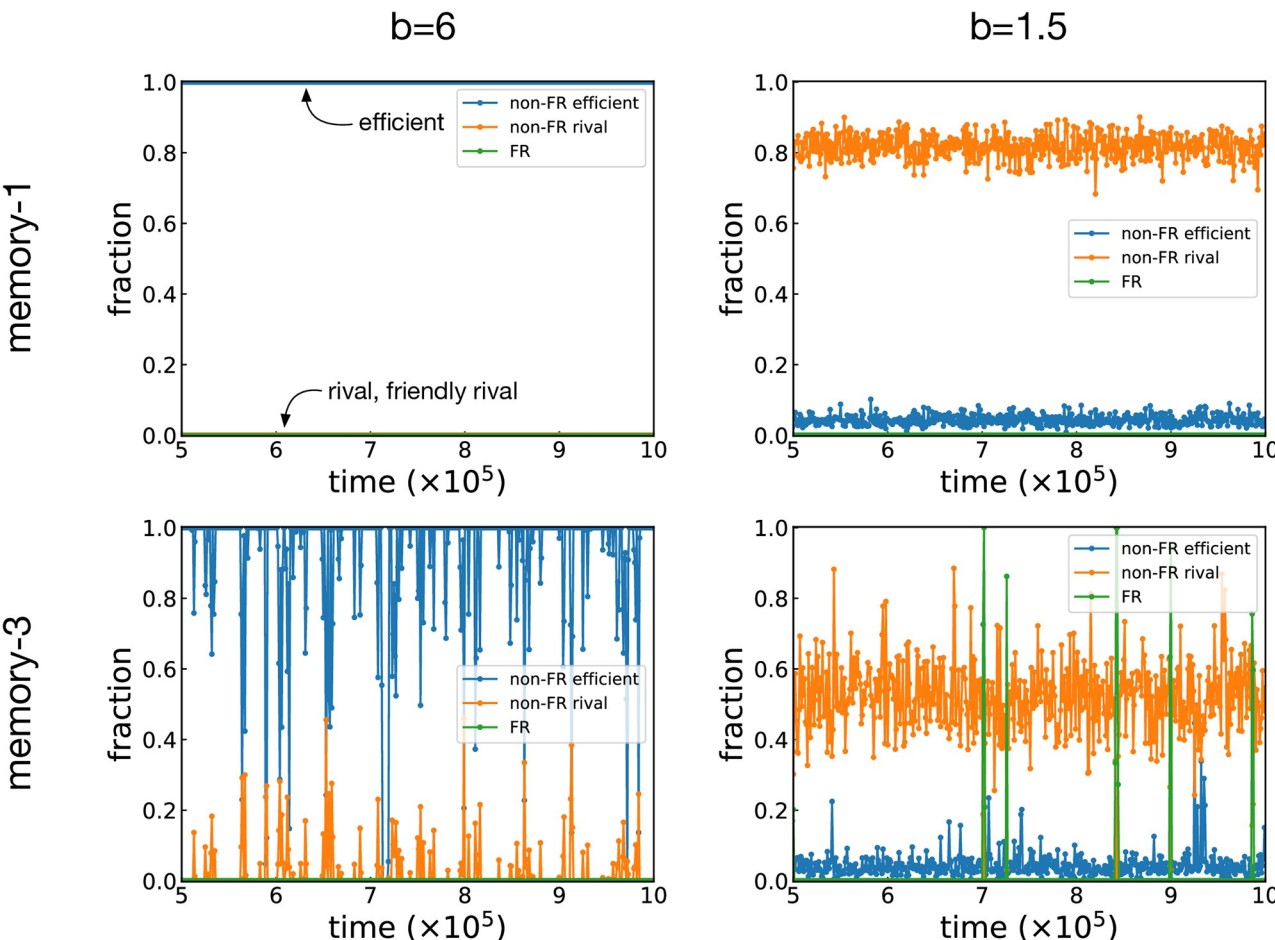

**Fig 5. Time series for the well-mixed populations.** Typical time series of the fractions of the non-FR efficient, non-FR rival, and FR strategies in the well-mixed populations. The top and bottom panels show the results for $\mathcal{S}(1)$ and $\mathcal{S}(3)$, respectively. The population size is $N = 64$, and the other simulation parameters are the same as in Fig 4. For the sake of better visualization, each time series plots every thousandth data point.

AllD within the group. The result is that efficient strategies keep wandering among groups and failing to conquer any of them. Note that both large- and small-$N$ effects can thus be experienced in this group-structured population.

As soon as FRs become available in $\mathcal{S}(3)$, they can survive both the in-group and out-group dynamics. Fig 8b illustrates a typical competition between an FR strategy and AllD. Because of the efficiency of the FR strategy, they are more likely to be imitated by AllD players via out-group imitations. The newly appeared FR player is also good at the intra-group selection because of its rivalry. Thus, it can take over the group and eventually the entire population. Once they enter the system, they are stable for a long time, as shown in Fig 7b. If residents have adopted an FR strategy, their evolutionary robustness assures that no mutant strategy has a fixation probability greater than $1/(NM)$. The greatest threat is a neutral drift process caused by non-FR efficient strategies cooperating with the residents. For instance, while CAPRI has a strictly higher payoff when pitted against AllC or WSLS, there are some non-FR efficient strategies that tie with CAPRI in $\mathcal{S}(3)$. These efficient strategies can thus replace FRs via nearly neutral drift. Indeed, Fig 7b shows that efficient strategies coexist with FRs to some extent while the rivals are almost entirely suppressed.

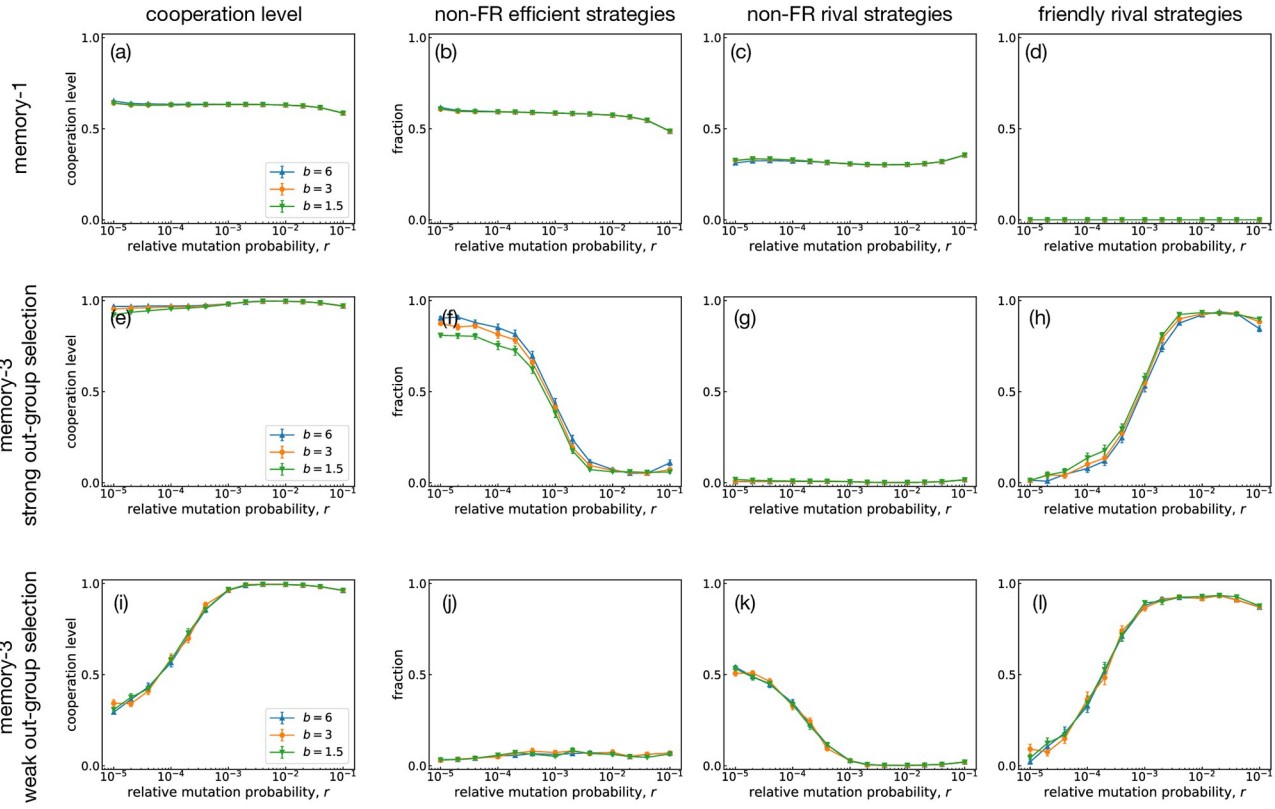

**Fig 6. Evolutionary simulations for the group-structured populations.** The number of groups is $M = 10^3$, and each group has $N = 2$. The panels on top (a-d) and middle (e-h) show the results for $\mathcal{S}(1)$ and $\mathcal{S}(3)$, respectively. The selection strength for out-group imitation is $\sigma_{out}$ $30/(b - 1)$. In the bottom panels (i-l), we show the results for $\mathcal{S}(3)$ with a weaker out-group selection strength $\sigma_{out}$ $3/(b - 1)$. From left to right, cooperation levels and the fractions of non-FR efficient strategies, non-FR rival strategies, and FR strategies are shown as functions of the relative mutation probability $r$. The error probability is $e = 10^{-6}$. Each data point has been obtained by averaging the results over $10^2$ independent runs.

The above argument also explains why a higher mutation rate stabilizes cooperation formed by FRs: it introduces rivals into the population, by which potentially threatening efficient strategies can be driven out. Fig 7c shows an example of time series in a group-structured population when $r$ is low. In the first half of this time series, an FR strategy occupies the majority,

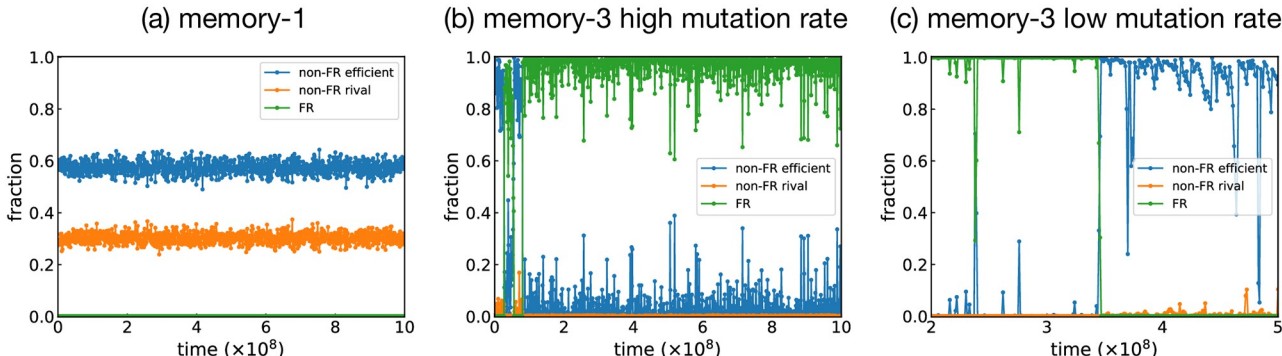

**Fig 7. Time series for the group-structured populations.** Typical time series showing the fractions of non-FR efficient, non-FR rival, and FR strategies. (a,b) $\mathcal{S}(1)$ and $\mathcal{S}(3)$ when the relative mutation probability is high ($r = 10^{-2}$). (c) An example of time series for $\mathcal{S}(3)$ with a low relative mutation probability ($r = 10^{-4}$), in which a non-FR efficient strategy replaces an FR strategy. The benefit of cooperation is $b = 3$, and the out-group selection strength is $\sigma_{out} = 15$. The other simulation parameters are the same as in Fig 6. For the sake of better visualization, each time series plots every millionth data point.

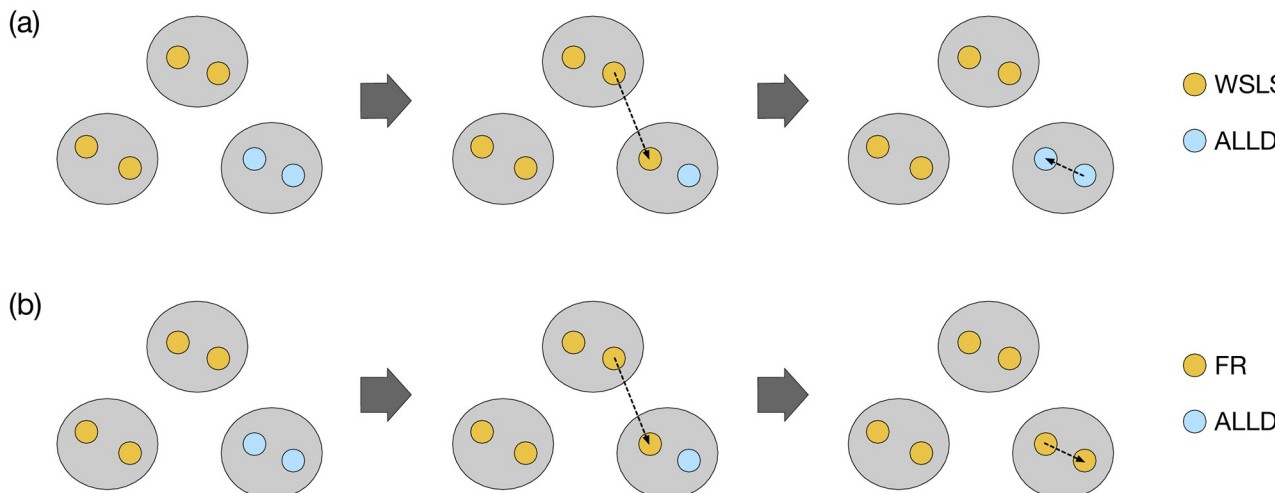

**Fig 8. Typical transitions in the group-structured populations.** (a) For $\mathcal{S}(1)$, WSLS and AllD often coexist in the population. An AllD player switches to WSLS via out-group imitation as WSLS players have higher payoffs. However, the newly appeared WSLS player is weak against AllD within the group, and it is almost surely replaced by AllD. Thus, the coexistence lasts for a long time. (b) For $\mathcal{S}(3)$, a different scenario is observed for FR strategies. After an AllD player switches to FR via out-group imitation, the FR player can resist AllD within the group. In this way, FRs can take over the entire population.

although it is invaded a few times by non-FR efficient strategies. At $t \approx 3.5 \times 10^8$, the FR strategy is replaced by an efficient one and thus wiped out from the population. Once this happens, the system is not as stable as in the first half, and we see rivals begin to rise. Because FRs are so rare, it will take a long time for another FR strategy to appear and settle down the situation. For FRs to survive long, therefore, $r$ needs to be high enough to suppress non-FR efficient strategies. Specifically, a rough guess would be that $r$ has to be greater than the fixation probability $\sim 1/M$ [28] that non-FR efficient strategies replace FRs through neutral selection. In S1 Appendix, we show the simulation results for $M = 100$. We find a crossover at the relative mutation rate $r \sim O(1/M)$, which is consistent with the above argument.

## Evolution of memory lengths

Finally, let us check how the memory lengths of strategies evolve. We measured the memory lengths $(m_1, m_2)$ of the resident strategy in each group and averaged these over the groups and over time. Fig 9 shows average memory lengths in well-mixed and group-structured populations. We have already seen that the evolutionary process depends on $b$ and $N$ in a well-mixed population. When the cooperation level is low, rivals with shorter memory lengths are the majority. More specifically, when $N$ or $b$ is low, the memory lengths are shorter than expected from the neutral case of $m_1 = m_2 = 1.5$. The opposite is true when $N$ and $b$ are high because the memory lengths become longer than the baseline. This observation is consistent with Fig 3b and 3c, which shows that rivals are easily found when memory lengths are small, compared with efficient strategies. A similar trend is also observed for a group-structured population [Fig 9b]: The cooperation level is positively correlated with the average memory lengths. The tendency is particularly striking when $m_2$ approaches three as $r$ increases. More interestingly, there is a notable difference between $m_1$ and $m_2$: We observe $m_2 > m_1$, which implies that FRs tend to memorize the opponent's history better than their own history of moves. It is also consistent with Fig 3d, according to which a greater number of FRs exist when $m_2 > m_1$.

### (a) well-mixed population

### (b) group-structured population

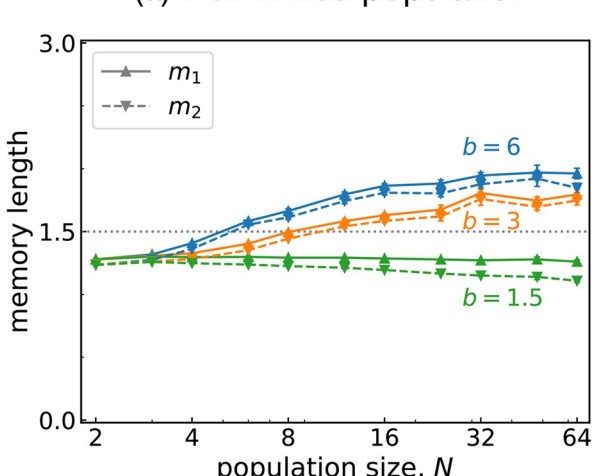
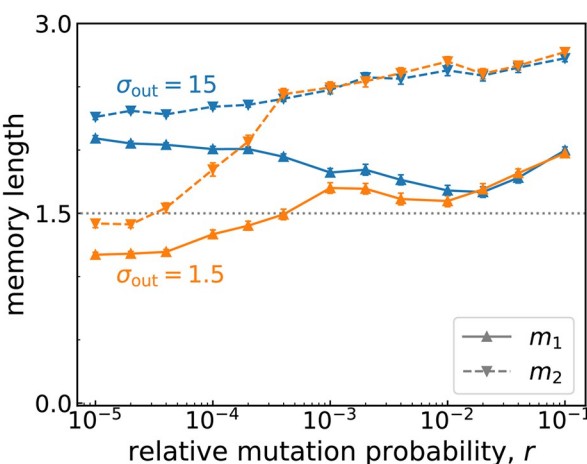

**Fig 9. Evolution of the memory lengths.** The average memory lengths ($m_1$, $m_2$) of the evolved strategies in $\mathcal{S}(3)$. (a) A well-mixed population and (b) a group-structured one show different behavior. The baselines $m_1 = m_2 = 3/2$ are depicted as dotted horizontal lines. The simulation parameters are the same as in Fig 6 unless otherwise mentioned. We used $b = 3$ in (b).

## Summary and discussion

In this paper, we have studied the evolutionary dynamics of well-mixed and group-structured populations in memory-1 and memory-3 strategy spaces. Our result demonstrates that group structure is an essential factor in manifesting the effects of memory. In group-structured populations, a strategy must succeed in both in-group and out-group selection processes, but their requirements are conflicting: In-group selection requires a strategy not to be beaten by its co-players, whereas out-group selection favors self-cooperative strategies. Since these conflicting demands for survival can be accommodated only by FRs, a group-structured population leads to a drastically different evolutionary consequence between $\mathcal{S}(1)$ and $\mathcal{S}(3)$. Namely, when only $\mathcal{S}(1)$ is available, we see stable coexistence between WSLS and AllD (or GRIM) irrespective of $b$, which is consistent with our previous study [28]. When the strategy space expands to $\mathcal{S}(3)$, by contrast, FRs prevail at $r \gtrsim O(1/M)$ with a cooperation level $\approx 100\%$ even for low $b$ (Fig 6). Whereas group structure has often been considered in the context of multilevel selection [42], our work proposes another use of it. An optimal condition for FRs is provided by creating different selection pressure depending on whether competition occurs within a group or between groups [43]. Once an FR strategy is adopted, the population plays a cooperative Nash equilibrium [10] with evolutionary robustness [12], combining cooperation and competition in a productive way [44].

In a well-mixed population, memory makes few differences in evolutionary trajectories whether we consider $\mathcal{S}(1)$ or $\mathcal{S}(3)$: The cooperation level is high because of the proliferation of efficient strategies when $b$ and $N$ are high, and rivals exhibit a low cooperation level otherwise, as has been reported previously [4, 23]. Although $\mathcal{S}(3)$ does contain FRs, they cease to play a pivotal role in a well-mixed population because they are easily outnumbered either by efficient strategies or by rivals depending on the environmental condition. Although FRs are observed more frequently than expected from random sampling, it is not enough to compensate for their small number, as shown in Fig 4h. Thus, it is hard to form cooperation in well-mixed populations for low $b$, even when complex strategies using additional memory lengths are available. By introducing group structure, we can see that nearly full cooperation is established because of FRs in the longer-memory strategy space.

Interestingly, a high relative mutation rate $r$ contributes to the stability of FRs [Fig 7b], which may look strange because they would be challenged by mutants frequently. The reason is that frequent mutation suppresses non-FR efficient strategies, which could potentially replace FRs via neutral drift. FRs are invulnerable to various mutants, while non-FR strategies are often weak against some mutants. This invulnerability makes FRs more advantageous when diverse mutants may appear. The mutation rate $r \gtrsim O(10^{-2})$ for which FRs are selected might look unusually high, but we note that $r$ is the relative frequency compared to the inter-group imitation events. Also, note that cultural transmission experiences more frequent explorative "mutation" than those assumed in biological models [45, 46]. We could even argue that a large amount of uncertainty may arise when someone tries to learn a strategy by observation, which could also result in a high effective mutation rate [25].

We consider that these results for the group-structured population are robust regardless of the model details. First, the results for $\mathcal{S}(2)$ are qualitatively similar to those for $\mathcal{S}(3)$, as shown in Fig A in SI. As expected, the key difference arises whether the strategy space contains FR strategies or not. Second, the results are insensitive to the benefit-to-cost ratio as shown in Fig 6, because the win-lose relationship between strategies is the decisive factor [28]. Although only the donation games are studied in this paper, we believe that the results would be similar for general PD games. This is because whether a strategy is an FR or not is independent of the elementary payoff matrix [12] as long as mutual cooperation is socially optimal ($2R > T + S$). Third, while we study the pairwise imitation process in this study, we expect that similar results would be obtained for other strategy updating rules such as the birth-death process. FRs are selected because of the selection pressures imposed by in-group and out-group competitions. These advantages of FR strategies are independent of the strategy updating rule. Overall, we expect the evolutionary consequences would be similar as long as the population has a group structure.

A possible model extension for future studies is the introduction of mixed strategies. FR strategies have measure zero in the mixed-strategy space because some prescriptions must be deterministic. For instance, $C$ must be played at mutual cooperation to be a partner and $D$ must be played at mutual defection to be a rival. Thus, it is hard to find FR strategies among mixed strategies. However, we still surmise that strategies that are close to FRs in the mixed-strategy space would be selected in a group-structured population although how to characterize the closeness to FRs remains a challenge. Another interesting model extension is the introduction of a "gradual" mutation to the model. We draw a mutant from the strategy space independently of the resident species in this study. For instance, a memory-3 mutant may suddenly appear in an AllD community. It could be more realistic to assume that a mutant is close to the resident species. Such a restriction on mutant strategies can alter the evolutionary dynamics [47] and would allow us to study the evolutionary path.

Even if group structure provides a favorable environment for FRs, one of the natural questions left in this study is how such structure emerges in the first place. It could be a matter of biology as is the case of *Dictyostelium discoideum* [48], but it can also be spontaneously induced by co-evolutionary network dynamics of interacting agents playing the PD game [49, 50]. The generality of this co-evolutionary mechanism implies that it can be ubiquitous across many different scales in society. It would be an interesting future direction to investigate the co-evolutionary dynamics of FRs and population structure.

We may think of FRs in terms of the emergence of other-regarding preference [51–54] in the sense that selection can favor an FR that compares its own payoff with the other player's. The existence of other-regarding is an interesting question because, in classical game theory, every player is assumed to care only about his or her own payoff. As we see in behavioral experiments and everyday experiences, by contrast, people often manifest other-regarding

preferences known as 'inequity aversion' [51, 55]. In our model, FR players express 'disadvantageous-inequity aversion' in the sense that they never let their co-players have higher payoffs whereas they do not care as long as their own payoffs are higher. Such a preference spontaneously emerges by playing FR strategies, while each player tries to increase his or her own payoff through imitation [see Eqs (4) and (5)]. Another recent study [56] also shows that selection favors learning rules that incorporate other-regarding preferences than selfish learning. These findings may help to understand the origin of other-regarding preferences in human society. Note also that the relation between other-regarding preference and group structure via FRs differs from the conventional idea that associates social preference with group selection [57, 58]: In our model, groups do not compete directly as is often assumed in the group-selection literature [42], and we do not view other-regarding preference as necessarily prosocial [59, 60], especially when it takes the form of rivalry. It is worth noting that the other-regarding preferences for partnership and rivalry lead to a refinement of Nash equilibrium, which still makes total sense among self-interested players.

Our study has also given theoretical predictions on the evolution of memory lengths. As shown in Fig 3, rivals can work well with short memory lengths, whereas cooperation seems to require a higher cognitive capacity. In particular, a large fraction of FRs exists when $m_2 \geq m_1 \geq 2$, meaning that an FR player has to remember the co-player's history better than his or her own. For this reason, the average memory length tends to be low (high) in environmental conditions where non-FR rivals (FRs) are favored. Recent studies of learning dynamics between two players predict a 'memory dilemma' in the sense that cooperative strategies with long memory lengths are invaded by simpler, less cooperative strategies [26, 61]. That is not inconsistent with our result, according to which rivalry will be favored in such a small population of two players. So far, our observation seems to be in qualitative agreement with previous studies that cooperative strategies with longer memory lengths will evolve [16, 17]. However, if we had an even larger strategy space, whether the average memory length in use would keep increasing [16, 62] is an open question. While there is a theoretical minimum memory length $m = 2$ to construct FR strategies for the IPD [11], the density of FRs will become extremely low as the memory length grows even longer [see Fig 3d]. If this trend continues, it is practically impossible to discover sophisticated FRs through random mutation from an even longer-memory strategy space. Therefore, as far as a two-person game is considered, there seem to be few reasons to go beyond $m = 3$. A recent experimental study also suggests that the optimal cooperation level occurs when the memory length is around $m = 2$ [63]. On the other hand, long memory may help to identify defectors [16], outperform a wide spectrum of strategies [12], and even reduce the cognitive load by providing a simple generalization [13, 14, 64], although such a complexity cost has not been incorporated in our model. Indeed, a previous simulation study shows that memory length gets continuously longer without the cost of memory capacity [16]. Testing these theoretical predictions and hypotheses with behavioral experiments will deepen our understanding of how much cognitive capacity is required in direct reciprocity [63, 65–70].

## Methods

### Calculation of long-term payoffs and cooperation levels

In general, strategies for the IPD need to define which action has to be taken after any history of previous interactions. Among infinitely many possible strategies, we focus on those with limited memory lengths. A well-known example is memory-one strategies, which condition their decision on the previous round. The relevant set of history profiles is {*CC, CD, DC, DD*}, where the first and second letters refer to the focal player's and the co-player's last actions,

respectively. Thus, a memory-one strategy can be represented as a 4-tuple,

$$\mathbf{p} = (p_{CC}, p_{CD}, p_{DC}, p_{DD}), \tag{10}$$

where $p_{ij}$ represents the player's cooperation probability for each given history profile $(i, j)$ from the previous round. We focus on deterministic strategies by setting $p_{ij}$ to either zero or one. The total number of memory-one deterministic strategies is therefore $2^4 = 16$.

A player may defect despite the intention to cooperate with probability $e \ll 1$ and vice versa. As a result, instead of the original strategy $\mathbf{p}$, the player effectively plays $(1 - e)\mathbf{p} + e\bar{\mathbf{p}}$, where $\bar{\mathbf{p}}$ is a vector with elements $\bar{p}_{ij} := 1 - p_{ij}$ for $i, j \in \{C, D\}$. When both players adopt memory-one strategies, the game is represented as a Markov chain, from which one can explicitly compute their payoffs and the cooperation levels. The states of this Markov chain are the possible outcomes of each round. When the players' (effective) strategies $\mathbf{p} = (p_{CC}, p_{CD}, p_{DC}, p_{DD})$ and $\mathbf{q} = (q_{CC}, q_{CD}, q_{DC}, q_{DD})$ are given, the transition matrix $T$ of the Markov chain takes the following form:

$$T = \begin{pmatrix} p_{CC} \cdot q_{CC} & p_{CC} \cdot \bar{q}_{CC} & \bar{p}_{CC} \cdot q_{CC} & \bar{p}_{CC} \cdot \bar{q}_{CC} \\ p_{CD} \cdot q_{DC} & p_{CD} \cdot \bar{q}_{DC} & \bar{p}_{CD} \cdot q_{DC} & \bar{p}_{CD} \cdot \bar{q}_{DC} \\ p_{DC} \cdot q_{CD} & p_{DC} \cdot \bar{q}_{CD} & \bar{p}_{DC} \cdot q_{CD} & \bar{p}_{DC} \cdot \bar{q}_{CD} \\ p_{DD} \cdot q_{DD} & p_{DD} \cdot \bar{q}_{DD} & \bar{p}_{DD} \cdot q_{DD} & \bar{p}_{DD} \cdot \bar{q}_{DD} \end{pmatrix}, \tag{11}$$

where $\bar{p}_{ij} := 1 - p_{ij}$ and $\bar{q}_{ij} := 1 - q_{ij}$ for $i, j \in \{C, D\}$. If $e > 0$, according to the Theorem of Perron-Frobenius, $T$ has a unique invariant distribution $\mathbf{v} = (v_{CC}, v_{CD}, v_{DC}, v_{DD})$. In particular, the $\mathbf{p}$-player's average cooperation level is $\gamma_{\mathbf{p},\mathbf{q}} := v_{CC} + v_{CD}$ whereas the $\mathbf{q}$-player's cooperation level is $\gamma_{\mathbf{q},\mathbf{p}} := v_{CC} + v_{DC}$. Consequently, the $\mathbf{p}$-player's long-term average payoff is given by

$$\pi_{\mathbf{p},\mathbf{q}} = b \cdot \gamma_{\mathbf{q},\mathbf{p}} - c \cdot \gamma_{\mathbf{p},\mathbf{q}}. \tag{12}$$

It is straightforward to extend this method to longer memory strategies. Memory-three strategies determine their subsequent actions based on the previous three rounds of interaction. Let us denote the relevant history profile by six letters separated by a comma, such as $(a_3 a_2 a_1, b_3 b_2 b_1)$, where $a_t \in \{C, D\}$ refers to what the focal player did $t$ rounds before, and $b_t \in \{C, D\}$ means the co-player's. For instance, a history profile $(CCC, CCD)$ indicates that the focal player continued cooperation over the last three rounds whereas the co-player defected in the previous round. A memory-three strategy prescribes an action for each of the $2^6 = 64$ history profiles, so it is represented by a 64-tuple,

$$\mathbf{p} = (p_{CCC,CCC}, p_{CCC,CCD}, \ldots, p_{DDD,DDD}), \tag{13}$$

where each element $p_{a_3 a_2 a_1, b_3 b_2 b_1}$ represents the player's cooperation probability for the given history profile. Similarly to the memory-one case, we work with an effective strategy $(1 - e)\mathbf{p} + e\bar{\mathbf{p}}$ in the presence of implementation error with probability $e > 0$. The repeated game between $\mathbf{p}$ and $\mathbf{q}$ is now represented by a Markov chain of 64 states, and the transition

probability from $(a_3a_2a_1, b_3b_2b_1)$ to $(a'_3a'_2a'_1, b'_3b'_2b'_1)$ is written as follows:

$$T_{(a_3a_2a_1,b_3b_2b_1)\to(a'_3a'_2a'_1,b'_3b'_2b'_1)}$$

$$= \begin{cases} 0 & \text{if } a'_3 \neq a_2 \text{ or } a'_2 \neq a_1 \text{ or } b'_3 \neq b_2 \text{ or } b'_2 \neq b_1 \\ p_{a_3a_2a_1,b_3b_2b_1} \cdot q_{b_3b_2b_1,a_3a_2a_1} & \text{if } (a'_3a'_2a'_1, b'_3b'_2b'_1) = (a_2a_1C, b_2b_1C) \\ p_{a_3a_2a_1,b_3b_2b_1} \cdot \bar{q}_{b_3b_2b_1,a_3a_2a_1} & \text{if } (a'_3a'_2a'_1, b'_3b'_2b'_1) = (a_2a_1C, b_2b_1D) \\ \bar{p}_{a_3a_2a_1,b_3b_2b_1} \cdot q_{b_3b_2b_1,a_3a_2a_1} & \text{if } (a'_3a'_2a'_1, b'_3b'_2b'_1) = (a_2a_1D, b_2b_1C) \\ \bar{p}_{a_3a_2a_1,b_3b_2b_1} \cdot \bar{q}_{b_3b_2b_1,a_3a_2a_1} & \text{if } (a'_3a'_2a'_1, b'_3b'_2b'_1) = (a_2a_1D, b_2b_1D) \end{cases}$$

(14)

Again, as a non-negative, irreducible, and aperiodic matrix, $T$ has a unique invariant distribution $\mathbf{v}$, from which the cooperation level of the $\mathbf{p}$-player is calculated as

$$\gamma_{\mathbf{p},\mathbf{q}} := \sum_{a_3,a_2,b_3,b_2,b_1} v_{a_3a_2C,b_3b_2b_1}.$$

(15)

## Memory length of a strategy

As already mentioned, strategies of $(m_1, m_2)$, where $m_1 \leq m$ and $m_2 \leq m$, constitute the memory-$m$ strategy space. It means that the set of memory-$m$ strategies includes strategies with shorter memory lengths as special cases. For instance, memory-one strategy space includes the so-called reactive memory-one strategies that condition the action on the co-player's previous action but not on its own. These strategies have $p_{CC} = p_{DC}$ and $p_{CD} = p_{DD}$ in common, we can say that $m_1 = 0$ in this case. Similarly, those with $p_{CC} = p_{CD}$ and $p_{DC} = p_{DD}$ can be said to have $m_2 = 0$ because they are indifferent to the co-player's history. If both $m_1$ and $m_2$ are zero, the strategies unconditionally have $p_{CC} = p_{CD} = p_{DC} = p_{DD}$.

In general, we calculate $(m_1, m_2)$ for a strategy represented by Eq (13) in the following way:

1. If there exists $(a_2, a_1, b_3, b_2, b_1)$ such that $p_{Ca_2a_1,b_3b_2b_1} \neq p_{Da_2a_1,b_3b_2b_1}$, it has $m_1 = 3$.

2. Else if there exists $(a_1, b_3, b_2, b_1)$ such that $p_{*Ca_1,b_3b_2b_1} \neq p_{*Da_1,b_3b_2b_1}$, where * denotes a wildcard, it has $m_1 = 2$.

3. Else if there exists $(b_3, b_2, b_1)$ such that $p_{**C,b_3b_2b_1} \neq p_{**D,b_3b_2b_1}$, it has $m_1 = 1$.

4. Otherwise, $m_1 = 0$.

   A similar algorithm is used for calculating $m_2$ as well.

## Judging efficiency and rivalry

When a strategy $\mathbf{p}$ is given, it is straightforward to judge its efficiency: It is an efficient strategy if $\lim_{e\to 0} \gamma_{\mathbf{p},\mathbf{p}} = 1$. Numerically, we judge efficiency if $\gamma_{\mathbf{p},\mathbf{p}}$ for $e = 10^{-4}$ is greater than 0.99. We have confirmed that the judgment is not sensitive to the threshold value. Another way of judgement is to use a graph-theoretical method [12], which checks probability currents with adding transitions of probability $O(e^k)$ systematically ($k = 0, 1, 2, \ldots$). We have compared the linear-algebraic and graph-theoretical methods with various strategies and verified their consistency. We also note that whether a strategy is a partner depends on $b$, whereas efficiency is independent of $b$, the benefit of cooperation. This is one of the reasons why we mainly work with efficiency in this paper.

To judge rivalry, we use a method based on the Floyd-Warshall algorithm [11, 12, 14, 71]. The idea can be explained as follows: A strategy $\mathbf{p}$ is a rival if it ensures that its co-player cannot obtain a higher long-term payoff regardless of the co-player's strategy as well as the initial state, when error probability $e$ is zero. We emphasize that the statement must be true even if the co-player's strategy has a long memory and/or if the strategy $\mathbf{p}$ is known to the co-player. One can judge the criterion by constructing a directed weighted graph $G(\mathbf{p})$. Consider the graph $G$ for $\mathbf{p}$ = TFT as an example. As a memory-one strategy, its action depends only on the the last round, so the relevant states are $CC$, $CD$, $DC$, and $DD$, each of which is represented as a node in $G$. We represent possible transitions among these four nodes as directed edges. For instance, at $CC$, TFT prescribes C, so the subsequent state is either $CC$ or $CD$. Each edge is assigned a weight corresponding to the relative payoff difference between the players. In our example, the self-edge from $CC$ to $CC$ thus has weight zero, whereas the edge from $CC$ to $CD$ has $-1$. In this way, we construct $G(\mathbf{p})$. The point is that only cycles in $G$ can contribute to the long-term payoff. Let a negative cycle denote a cycle along which the total sum of weights is negative. If a negative cycle exists, the co-player can take advantage of it to exploit the focal player. Conversely, the absence of such negative cycles in $G(\mathbf{p})$ guarantees that no strategy can obtain a higher long-term payoff than $\mathbf{p}$. The presence of a negative cycle in a graph can be detected by the Floyd-Warshall algorithm in polynomial time, and this method is straightforwardly extensible to longer-memory strategies [12].

The numbers of strategies in Fig 3 are obtained in the following way: When $m_1 + m_2 \leq 4$, we can enumerate all the possible strategies and check their efficiency and rivalry one by one. This enumeration approach becomes impractical when $m_1 + m_2 > 4$, so we have estimated the fraction of efficient ones and that of rival ones from randomly sampled $10^6$ strategies. As for FR strategies, it is possible to directly obtain the complete list of FR strategies using the algorithm proposed in [12] because they are infrequent. We obtained the exact number of FR strategies instead of Monte Carlo sampling.

## Monte Carlo simulations

The Monte Carlo simulations for well-mixed populations have been conducted as follows. We assume the limit of a low mutation rate, in which at most one mutant can compete with the resident strategy, and no other mutation occurs until this mutant takes over the population or dies out. At each time step, a mutant strategy $Y$ is randomly sampled according to the two-step process [Eq (9)], and it replaces the resident strategy $X$ with fixation probability $\rho_{X \to Y}$ [Eq (6)] [72]. We iterate this process for $10^6$ time steps with discarding the initial $10^5$ steps. The cooperation level in Fig 4a is calculated as the time average of the cooperation levels of the resident strategies, given by Eq (15).

For the simulations of group-structured populations, we assume that intra-group dynamics is fast enough compared to inter-group dynamics and mutation, i.e., $\mu_{\text{in}} \gg \mu_{\text{out}}$ and $\mu_{\text{in}} \gg \nu$. As we have assumed in the case of well-mixed populations, a group is usually occupied by a single resident strategy, which can be replaced by a different one that appears through either mutation or out-group imitation and succeeds in fixation. The Monte Carlo simulations have been conducted as follows.

1. Prepare a set of $M$ randomly selected strategies as the initial state.

2. Choose one of the $M$ groups randomly. Let us denote the strategy of this group as $X$.

3. With probability $r$, the group undergoes mutation.

   a. Introduce a mutant strategy $Y$ according to the two-step sampling scheme [Eq (9)].

 b. Replace $X$ by $Y$ with fixation probability $\rho_{X \rightarrow Y}$ [Eq (6)].

4. With probability $1 - r$, the group undergoes out-group imitation.

 a. Choose randomly one of the other groups. Let $Y$ denote its strategy.

 b. Replace $X$ by $Y$ with probability $T_{X \rightarrow Y}$ [Eq (7)].

5. Go back to step 2.

 The above process is repeated for $10^9$ steps, from which we discard the initial $10^8$ steps. We have used OACIS and CARAVAN to manage the simulation results [73, 74].

## Supporting information

**S1 Appendix. Simulations for other settings.** The simulation results for $\mathcal{S}(2)$ and for different parameters are shown. In addition, the model with the full separation of time scale is studied. **Fig A. Evolutionary simulations for memory-2 strategy space.** Evolutionary simulations for $\mathcal{S}(2)$ in a group-structured population ($M = 10^3$, $N = 2$). The upper (a-d) and the lower (e-h) show the results for different out-group selection strengths $\sigma_{\text{out}} = 30/(b - 1)$ and $\sigma_{\text{out}} = 3/(b - 1)$, respectively. The other simulation parameters are the same as those in Fig 6. **Fig B. Simulations with different parameters.** Effects of $M = 10^2$, to be compared with Fig 6. The other parameters are $N = 2$, $e = 10^{-6}$ and $\sigma_{\text{in}} = 30/(b - 1)$. For panels (a-h), we have used $\sigma_{\text{out}} = 30/(b - 1)$ whereas $\sigma_{\text{out}} = 3/(b - 1)$ for the bottom panels. Each simulation runs for $10^8$ time steps, and the results are averaged over 10 independent runs. **Fig C. Simulations with completely separated time scales.** The simulation results for the case where the timescales for out-group imitations and mutations are completely separated. From left to right, the panels show the cooperation level, (b) the fractions of non-FR efficient strategies, (c) the fractions of non-FR rival strategies, and (d) the fractions of FR strategies. The parameters are same as those in Fig 6a–6f: $M = 10^3$, $N = 2$, $e = 10^{-6}$, and $\sigma_{\text{in}} = \sigma_{\text{out}} = 30/(b - 1)$. Simulations were conducted for $10^6$ steps, discarding the initialization period of $10^5$ time steps, and the results were averaged over 10 independent runs.
(PDF)

## Acknowledgments

The authors sincerely thank C. Hilbe for his careful reading and valuable comments on the manuscript. Y.M. and S.K.B. appreciate the APCTP for its hospitality during the completion of this work. Part of the results is obtained by using the Fugaku computer at RIKEN Center for Computational Science (Proposal number ra000002).

## Author Contributions

**Conceptualization:** Yohsuke Murase.

**Formal analysis:** Yohsuke Murase.

**Funding acquisition:** Yohsuke Murase, Seung Ki Baek.

**Investigation:** Yohsuke Murase.

**Methodology:** Yohsuke Murase, Seung Ki Baek.

**Software:** Yohsuke Murase.

**Validation:** Seung Ki Baek.

**Writing – original draft:** Yohsuke Murase.

**Writing – review & editing:** Yohsuke Murase, Seung Ki Baek.

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
