## [Decision Letter · Decision Letter 0]

16 Apr 2023

Dear Dr. Murase,

Thank you very much for submitting your manuscript "Grouping promotes both partnership and rivalry with long memory in direct reciprocity" for consideration at PLOS Computational Biology.

As with all papers reviewed by the journal, your manuscript was reviewed by members of the editorial board and by several independent reviewers. In light of the reviews (below this email), we would like to invite the resubmission of a significantly-revised version that takes into account the reviewers' comments.

All of the reviewers found the paper to be interesting and suitable for publication. However, they raised important points that need to be addressed in revisions. Specifically, the reviewers highlighted the sensitivity of the results to the relative rate of mutation to out-group imitation and the need to consider pure vs. mixed strategies in the simulations. These issues must be thoroughly addressed before publication.

We cannot make any decision about publication until we have seen the revised manuscript and your response to the reviewers' comments. Your revised manuscript is also likely to be sent to reviewers for further evaluation.

Sincerely,

Feng Fu

Academic Editor

PLOS Computational Biology

James O'Dwyer

Section Editor

PLOS Computational Biology

All of the reviewers found the paper to be interesting and suitable for publication. However, they raised important points that need to be addressed in revisions. Specifically, the reviewers highlighted the sensitivity of the results to the relative rate of mutation to out-group imitation and the need to consider pure vs. mixed strategies in the simulations. These issues must be thoroughly addressed before publication.

Reviewer's Responses to Questions

**Comments to the Authors:**

Reviewer #1: In line with Zero-determinant (ZD) strategy, this work underlines the so-called direct reciprocity. The authors presumed Donor & Recipient (D & R) game, where T = b, R = b – c, P = 0, and S = - c. They further presume c=1 without any loss of generality, which makes Chechen-type dilemma Dg’ := (T – R) / (R – P) = 1/(b-1) & Stag Hunt-type dilemma Dr’ := (P – S) / (R – P) = 1/(b-1), i.e, Dg’ = Dr’ (that’s why D&R game is ensured). A finite and well-mixed population is premised and PW-Fermi is applied.

Their main concern is whether a well-mixed or group-structure treatment brining any significant result on reciprocity, and also memory effect delivering any difference.

By going thru a stringent MAS (by taking Monte Caro simulation) approach with support of some theoretical underline, the authors successfully obtained some new findings, which can be highly evaluated and quite informative to be shred to the audience in the scientific arena.

Scientific health and solidness seem reliable.

Hence, I would like to recommend this work to be published.

Just one technical suggestion to add impressiveness is given as below.

As I abovementioned, the game structure the authors presumed belong to D & R game, one of sub-classes of PD, which is a standard template for PD theoretical biologists have heavily favored. That’s fine. In usual definition there are two parameters; b and c. They presumed c=1 unity, which is perfectly ok. But they should carefully explain it, and should mention that the dilemma strength in such case; measured by the universal dilemma strength by Dg’ and Dr’, is parameterized by a single parameter; b (more precisely; 1/(b-1)). The additional part should be added to Model depiction, which should be accompanied by citation and review on relevant literatures, for instance, (i) Relationship between dilemma occurrence and the existence of a weakly dominant strategy in a two-player symmetric game, BioSystems 90(1), 105-114, 2007, (ii) Universal scaling for the dilemma strength in evolutionary games, Physics of Life Reviews 14, 1-30, 2015, (iii) Scaling the phase- planes of social dilemma strengths shows game-class changes in the five rules governing the evolution of cooperation, Royal Society Open Science, 181085, 2018, (iv) Sociophysics Approach to Epidemics, Springer, 2021.

Reviewer #2: In this paper “Grouping promotes both partners and rivalry with long memory in direct reciprocity,” the authors investigate the relationship between a population’s structure and the memory length that emerges in a repeated donation game. As expected, memory length plays little role in unstructured populations, where defection dominates. It is in group-structured populations that successful memory-3 strategies start to emerge, and among those the successful ones have a much greater abundance than if randomly sampled from the strategy space. These “friendly rivals” play two roles, essentially. They ensure that competition within a group is rivalrous, while competition between groups involves self-cooperating strategies. These two properties cannot be satisfied in the simpler class of memory-one strategies, and the authors show that they do exist in higher-memory spaces (although they are still rare).

I really enjoyed reading this paper. The authors do an excellent job of presenting the background and motivation for their study. While I do have some comments below, I think this paper should be published in PLOS Computational Biology after some minor revisions.

Abstract: remove “large-scale”

Abstract: it’s not clear why there is a jump from memory-one to memory-three (why not memory-two, a reader might ask?)

Introduction, paragraph two: I would not say that it reduces a two-body problem to a one-body problem, since this could be said of any play in which one agent fixes a strategy and allows the other to choose. Instead, as other ZD studies have stated, it would be more descriptive to say that it induces an ultimatum on the other player.

Introduction, paragraph four: When mentioning CAPRI, you should discuss briefly the finding that if a player uses a memory-one strategy then they can assume WLOG that the opponent also uses a memory-one strategy. At first, it seems confusing that a longer-memory strategy could outperform a memory-one strategy, but in the background, I assume what the authors are referring to is the (population-based) performance of memory-k versus memory-k for k>1?

Figure 1: change the shading of the feasible region to another color or change the blue bubble above to something else. It is confusing when the colors are referred to in the text because they look the same to the naked eye (although maybe they are slightly different shades?).

Introduction, final paragraph: remove “large-scale”

Introduction, final paragraph: oh, so you mean boundary strategies within the memory-k (k in {1,3}) space, since you state that they have finite cardinality?

Model, paragraph one: is the purpose of implementation errors just to ensure that the Markov chain is ergodic?

Before equations (3) and (4), \\pi_X and \\pi_Y need to be defined. In a finite group, does an individual interact with everyone and average those payoffs to get \\pi_X and \\pi_Y?

On line 198, replace “naively” with “uniformly”

Remove “significant” from line 203 to avoid confusion with statistical significance.

Right at the beginning of “Results” the figure is referred to for a qualitative comparison between memory-1 and memory-3 strategies. What is the “fraction” that appears there? Is this a stationary fraction after many steps? Or up to some finite generation?

On line 231, remove “Nash” since a SPE is a refinement of a Nash equilibrium.

On line 237, what is an example of a “dangerous mutant” from S(3) that can threaten WSLS? Is there an intuitive reason for why WSLS is susceptible strategies with longer memory capacities?

When talking about “evolution of memory lengths,” the authors mention that m_2 > m_1 is favored, meaning players care more about having a longer memory of the opponent than of themselves. Could one reason for this be the fact that the central game considered is a donation game (which is additive)? Would the same qualitative findings hold for the non-additive PD with (R,S,T,P) = (3,0,5,1)?

One weakness of this study, which the authors mention indirectly toward the end, is that it relies on imitation with mutation. Since strategies here are not binary actions which can be easily observed, the ability to imitate a strategy presupposes that these strategies can be observed. I don’t really think that just raising the mutation rate accounts for this, at least in terms of human behavior. Mutation rates at the level of the conditional strategy are more natural in non-cultural settings like birth-death processes. If one were to take a group-structured population with BD updating a migration, would you observe similar results? I am not suggesting the authors add a lot of new material on BD updating, but it would be really helpful for the paper if this could be checked in some level of depth and at least commented on in the paper.

Reviewer #3: This paper explores the evolution of "friendly-rival" (FR) strategies in well-mixed and group structured populations playing n=3 public goods games. FR strategies are of interest because 1) they ensure mutual cooperation against self but also ensure a strictly greater payoff against any player that deviates from mutual cooperation and 2) they require memory m>1. They are thus qualitatively different than they types of strategies commonly studied in iterated games which have memory m=1 (TFT, WSLS and so on). The authors show that FR strategies do not tend to dominate in well mixed populations (consistent with previous work), largely due to not being "findable" in a high dimensional strategy space associated with memory m=3 (which is the minimum memory length required for FR strategies in 3-player public goods games). However they show that in group structured populations in which imitation of out-groups is possible, FR strategies evolve and stabilize high-levels of cooperation.

This is a fascinating paper and I support publication. I have two comments that the authors might wish to address in a revision

1) The authors study the limit in which in-group imitation is much faster than both out-group imitation and mutation. They then study the impact of the parameter r=nu/(mu_out+nu) on the evolutionary dynamics (i.e. the relative rate of mutation to out-group imitation). This generates a kind of weak-mutation limit at the group level (each group will be composed of at most two strategies at any given time). In the Appendix they also study a full separation of timescales in which mutation is truly weak (i.e. at most two strategies exist in the population at any one time, and in-group imitation is fast compared to out-group). These are both interesting limits but it is not clear to me that mu_out<<mu_in a="" assumption.="" do="" how="" is="" look="" mu_out="" results="" sensible="" the="" when="">

2) In the simulations it's not entirely clear to me whether only pure strategies are considered. In Figure 3 and on page 7 pure strategies are discussed but in Figure 1 strategies such as extortion are discussed which use cooperation probabilities other than 0 or 1. This choice matters because the results in well-mixed populations are driven by the relative rarity of FR strategies. Do FR strategies become more or less rare when we move away from pure strategies, and how does that impact the results in group structured populations?

3) In the evolution of memory section (Figure 9), how are mutations that change memory length implemented? That is, do we just assume that any strategy of any memory length can mutate to any other? Or is there some sense in which mutations are local (eg memory can only increase by one unit at a time)?</mu_in>

**Have the authors made all data and (if applicable) computational code underlying the findings in their manuscript fully available?**

Reviewer #1: Yes

Reviewer #2: Yes

Reviewer #3: Yes

PLOS authors have the option to publish the peer review history of their article (what does this mean?). If published, this will include your full peer review and any attached files.

Reviewer #1: No

Reviewer #2: No

Reviewer #3: No
---

## [Decision Letter · Decision Letter 1]

30 May 2023

Dear Dr. Murase,

We are pleased to inform you that your manuscript 'Grouping promotes both partnership and rivalry with long memory in direct reciprocity' has been provisionally accepted for publication in PLOS Computational Biology.

Best regards,

Feng Fu

Academic Editor

PLOS Computational Biology

James O'Dwyer

Section Editor

PLOS Computational Biology

Reviewer's Responses to Questions

**Comments to the Authors:**

Reviewer #1: The MS with this revised version well replies the commnets given. Thus, I would suggest to acceptance...

Reviewer #2: I appreciate the authors’ responses to my concerns, and I am happy to recommend publication in PLOS Computational Biology.

Reviewer #3: The authors have addressed all my comments in the revised manuscript. I am happy to support publication

**Have the authors made all data and (if applicable) computational code underlying the findings in their manuscript fully available?**

Reviewer #1: Yes

Reviewer #2: None

Reviewer #3: Yes

PLOS authors have the option to publish the peer review history of their article (what does this mean?). If published, this will include your full peer review and any attached files.

Reviewer #1: No

Reviewer #2: No

Reviewer #3: No

---

## [Editor Report · Acceptance letter]

16 Jun 2023

PCOMPBIOL-D-23-00226R1 

Grouping promotes both partnership and rivalry with long memory in direct reciprocity

Dear Dr Murase,

I am pleased to inform you that your manuscript has been formally accepted for publication in PLOS Computational Biology. Your manuscript is now with our production department and you will be notified of the publication date in due course.

With kind regards,

Zsofia Freund
